# Identification of CD44 as a Reliable Biomarker for Glioblastoma Invasion: Based on Magnetic Resonance Imaging and Spectroscopic Analysis of 5-Aminolevulinic Acid Fluorescence

**DOI:** 10.3390/biomedicines11092369

**Published:** 2023-08-24

**Authors:** Akihiro Inoue, Takanori Ohnishi, Masahiro Nishikawa, Hideaki Watanabe, Kosuke Kusakabe, Mashio Taniwaki, Hajime Yano, Yoshihiro Ohtsuka, Shirabe Matsumoto, Satoshi Suehiro, Daisuke Yamashita, Seiji Shigekawa, Hisaaki Takahashi, Riko Kitazawa, Junya Tanaka, Takeharu Kunieda

**Affiliations:** 1Department of Neurosurgery, Ehime University School of Medicine, 454 Shitsukawa, Toon, Ehime 791-0295, Japan; ma.nishikawa1985@gmail.com (M.N.); whideaki@m.ehime-u.ac.jp (H.W.); kk_0145@yahoo.co.jp (K.K.); y.ohtsuka0818@gmail.com (Y.O.); shirabem13@gmail.com (S.M.); satoshi.suehiro@gmail.com (S.S.); yamadai551208@gmail.com (D.Y.); shigekaw@m.ehime-u.ac.jp (S.S.); kuny@m.ehime-u.ac.jp (T.K.); 2Department of Neurosurgery, Washoukai Sadamoto Hospital, 1-6-1 Takehara, Matsuyama, Ehime 790-0052, Japan; tohnishi@m.ehime-u.ac.jp; 3Division of Diagnostic Pathology, Ehime University Hospital, 454 Shitsukawa, Toon, Ehime 791-0295, Japan; hechota@gmail.com (M.T.); riko.kitazawa@gmail.com (R.K.); 4Department of Molecular and Cellular Physiology, Ehime University School of Medicine, 454 Shitsukawa, Toon, Ehime 791-0295, Japan; hajime-y@m.ehime-u.ac.jp (H.Y.); jtanaka@m.ehime-u.ac.jp (J.T.); 5Division of Pathophysiology, Faculty of Pharmaceutical Sciences, Hokuriku University, Kanazawa, Ishikawa 920-1181, Japan; h-takahashi@hokuriku-u.ac.jp

**Keywords:** glioblastoma, invasion, CD44, 5-aminolevulinic acid, fluorescence spectroscopy, glioma stem cell, biomarker

## Abstract

Recurrent glioblastoma multiforme (GBM) is largely attributed to peritumoral infiltration of tumor cells. As higher CD44 expression in the tumor periphery correlates with higher risk of GBM invasion, the present study analyzed the relationship between CD44 expression and magnetic resonance imaging (MRI)-based invasiveness of GBM on a large scale. We also quantitatively evaluated GBM invasion using 5-aminolevulinic acid (5-ALA) spectroscopy to investigate the relationship between CD44 expression and tumor invasiveness as evaluated by intraoperative 5-ALA intensity. Based on MRI, GBM was classified as high-invasive type in 28 patients and low-invasive type in 22 patients. High-invasive type expressed CD44 at a significantly higher level than low-invasive type and was associated with worse survival. To quantitatively analyze GBM invasiveness, the relationship between tumor density in the peritumoral area and the spectroscopic intensity of 5-ALA was investigated. Spectroscopy showed that the 5-ALA intensity of infiltrating tumor cells correlated with tumor density as represented by the Ki-67 staining index. No significant correlation between CD44 and degree of 5-ALA-based invasiveness of GBM was found, but invasiveness of GBM as evaluated by 5-ALA matched the classification from MRI in all except one case, indicating that CD44 expression at the GBM periphery could provide a reliable biomarker for invasiveness in GBM.

## 1. Introduction

Glioblastoma multiforme (GBM) is the most malignant brain tumor, characterized by a highly invasive nature and inter- and intratumor heterogeneity. These features make GBM a tumor with poor prognosis due to diffuse infiltration of tumor cells into normal peritumoral brain tissues. In particular, tumor cells existing in the peritumoral area, where gadolinium (Gd) enhancement is not detected on magnetic resonance imaging (MRI), create a big obstacle to radical excision of the tumor, resulting in infiltrating tumor cells being left in place. The current therapy in the primary GBM consists of maximal tumor resection with minimal brain injury, the extended local radiotherapy with 60 Gy, and the concomitant chemotherapy with temozolomide. In addition to non-radicality in tumor resection, various factors make GBM extremely poor tumors. These include the fact that GBM cells show a resistance to chemotherapy due to elevated extracellular exclusion of chemotherapeutic agents and the non-disrupted blood–brain barrier and a resistance to radiotherapy due to upregulation of repair enzymes for DNA damages besides a hypoxic environment surrounding tumor cells. Glioma stem-like cells (GSCs), which constitute a small population in GBM but have abilities to make self-proliferation, multilineage differentiation to neurons and astrocytes, and rapid tumorigenesis in a mouse xenograft model, could participate in all these forms of resistance to the therapy. If GSCs are included in the infiltrating tumor cells, they will become a key cause of early tumor recurrence for GBM [1,2,3,4,5,6]. Specifically, we found that GSCs expressing high cluster of differentiation (CD) 44, which were established by the primary culture of tumor cells obtained from the tumor periphery of highly CD44-expressing GBM patients, can present a high invasive feature [7,8].

CD44 is a transmembrane glycoprotein expressed in various cells, such as embryonic stem cells, differentiated cells and cancer cells, including GBM [9,10]. CD44 is a multi-functional receptor molecule with important roles not only in mediating cell-to-cell and cell-to-extracellular matrix interactions, but also in transducing intracellular signaling pathways that promote cell migration, invasion, proliferation, and tumorigenesis in malignant tumors (Appendix A) [11]. Our studies demonstrated that GBM with higher expression of CD44 in the tumor periphery than in the core correlated well with the highly invasive-type GBM as defined on magnetic resonance imaging (MRI), associated with poorer survival of patients. Conversely, lower expression of CD44 in the tumor periphery corresponded with low invasive-type GBM and longer survival [7]. In those studies, the ratio of CD44 expression in the tumor periphery to expression in the tumor core (P/C ratio) could be evaluated as a marker of GBM invasiveness. CD44 expression in the tumor periphery alone was not associated with the invasiveness of GBM. Why only the P/C ratio of CD44 expression correlated with GBM invasiveness remains unclear. To confirm the correlation between CD44 expression and invasiveness of GBM, the present study investigated in an increasing number of patients with GBM and analyzed whether CD44 offered the most reliable indicator for predicting GBM invasiveness.

Elucidating whether CD44 expression correlates with the degree of actual invasiveness of GBM apart from MRI is another critical issue. To achieve supra-total resection of malignant glioma, fluorescence-guided surgery (FGS) has recently been adopted for glioma surgery using image-guided neuro-navigation systems [6,12]. This technique has been developed to enhance the extent of GBM resection, and supra-total resection of GBM has been demonstrated to contribute to good patient prognosis [13,14]. Among the various photosensitizers available, 5-aminolevulinic acid (5-ALA), an amino acid and natural precursor in heme biosynthesis, has been widely used as a fluorescence-inducing molecule in intraoperative FGS for malignant glioma [15]. Many studies have reported that the intensity of fluorescence from 5-ALA-derived protoporphyrin IX (PpIX) in brain tumors correlates with tumor cell grade and malignancy, with both sensitivity and specificity exceeding 80% [6,16,17]. Fluorescence intensity of 5-ALA-derived PpIX is usually evaluated by visually assessed strength of red color. Tumors showing fluorescence with a strong red color correspond to malignant tumors with high grade (GBM). Tumors presenting almost no red color correspond to low-grade glioma, and tumors showing a pale red or pinkish color correspond to grade-III malignant glioma. Photodynamic diagnosis using 5-ALA appears useful to visualize tumor cells not recognizable under white light in the operative field, but such photodynamic diagnosis sometimes fails to detect tumor cells that do not show red-color fluorescence and cannot quantitatively evaluate the fluorescence intensity of tumor cells. To overcome these problems, the present study introduced spectroscopy for 5-ALA-derived PpIX fluorescence using a violet laser diode (VLD) to enhance the visualization of tumor cells and allow for quantitative evaluation of 5-ALA fluorescence intensity from infiltrating glioma cells. We evaluated the degree of invasiveness of GBM by analyzing the relationship between spectroscopic 5-ALA fluorescence signal intensity and tumor proliferation rate (Ki-67 staining index [SI]), which is considered to represent the density of infiltrating tumor cells in the peritumoral area of GBM [5]. We then investigated whether the P/C ratio of CD44 expression correlated with the degree of invasiveness of GBM as evaluated by spectroscopic 5-ALA fluorescence intensity. These studies might identify potential biomarkers for evaluating the degree of GBM invasiveness. Identification of a novel marker for GBM invasion will provide a useful tool for not only elucidating the molecular mechanisms underlying glioma invasion, but also developing new strategies for the effective treatment of GBM.

## 2. Materials and Methods

This study was approved by the Ethics Committee for Clinical Research of Ehime University Hospital (approval no. 2211011). All procedures were performed in accordance with the ethical standards of the 1964 Declaration of Helsinki and its later amendments.

### 2.1. Patients and Study Design

Fifty patients with histologically verified GBM treated under the same protocol in the Department of Neurosurgery at Ehime University Hospital between April 2014 and January 2021 were enrolled in the present study. Informed consent was obtained from each participant after receiving explanations of the potential risks of positron emission tomography (PET) and MRI, surgical procedure, radio-chemotherapy, and oral administration of 5-ALA in patients undergoing 5-ALA-guided surgery. Neurological findings were assessed preoperatively, immediately after surgery, and 3 months postoperatively. The same neurosurgeon and neuroradiologist analyzed the clinical records and radiological examinations of these patients on admission, within 72 h and 3 months after surgery. Other requirements for enrollment included: Karnofsky performance status (KPS) score ≥60; tumor resection more than biopsy (gross total resection (GTR), 100% resection of tumor volume; subtotal resection (STR), ≥95% but <100% resection of tumor volume; or partial resection (PR), <95% resection of tumor volume); and bevacizumab, a monoclonal antibody targeting vascular endothelial growth factor (VEGF), therapy at tumor recurrence. The treatment protocol comprised tumor resection followed by radiotherapy (60 Gy) and chemotherapy with temozolomide (TMZ), in accordance with the Stupp regimen [2]. Ki-67 SI was evaluated by immunohistochemistry and used to estimate the number of proliferative tumor cells in tumor tissue samples. In all 50 patients, mRNA expression of CD44 was examined in the periphery and core of the tumor, to clarify what patterns of CD44 expression correlated most significantly with GBM invasiveness. Then, among the 50 patients, the most recent 21 patients who had been operated on under the guidance of spectroscopic 5-ALA fluorescence were entered into the present study to quantitatively analyze the degree of GBM invasiveness. Finally, the correlation between CD44 expression in the tumor periphery and tumor invasiveness as evaluated by spectroscopic analysis of 5-ALA fluorescence was investigated.

### 2.2. Imaging Studies and Analysis

MRI examinations were achieved using a 3-T scanner (Achieva; Philips, Best, the Netherlands) with a standard head coil. Axial, coronal, and sagittal T1-weighted images were obtained in 2 mm slice thickness before and after intravenous administration of gadolinium-diethylenetriamine pentaacetic acid (0.1 mmol/kg). Based on the features observed on MRI, patients were divided into a high-invasive (HI) type and a low-invasive (LI) type. Four features were used to define the invasive type, including morphology of the tumor margin, enhancement of the tumor margin wall, peritumoral edema, and central necrosis (Table 1; Figure 1). In cases presenting equivocal features of invasiveness on MRI, methionine (Met)-PET was performed. Details of the criteria for these phenotypes in GBM on MRI and PET have been described previously [5].

### 2.3. Image-Guided Navigation Surgery and Selective Tissue Sampling

Tumor resection was performed by echo-linked navigation-guided microsurgery using MRI and Met-PET fusion images and fence-post catheter techniques. Tumor samples were separately obtained from the core and periphery of the tumor using surgical techniques, as previously described [7,8]. In addition, tumor tissues were obtained from several different areas in the GBM, including areas of SUVmax on Met-PET, and areas that showed positive 11C-Met uptake but no Gd-enhancement in the tumor periphery.

### 2.4. Immunohistochemistry

Tumor samples were fixed in buffered formalin and embedded in paraffin according to our regular histological examination. Hematoxylin and eosin (HE)-stained sections and immunohistochemical studies were obtained from all patients. In the immunohistochemical studies, deparaffinized 4 µm tissue sections from paraffin blocks were hydrated in a graded alcohol series and underwent heat-activated antigen retrieval. After blocking endogenous peroxidase activity, tissue was incubated with anti-Ki-67 mouse monoclonal antibody, clone MIB-1 code GA626 (1:400) (DAKO, Santa Clara, CA, USA). Sections were then washed and incubated with biotinylated secondary antibody for 30 min at room temperature. Reaction complexes were visualized with diaminobenzidine and counterstained with hematoxylin. Appropriate negative and positive control experiments were run for all antibodies tested.

### 2.5. RNA Isolation and Quantitative Real-Time Reverse Transcription Polymerase Chain Reaction (qRT-PCR)

Total RNA was extracted from both the core and periphery of each tumor sample using ISOGEN (Nippon Gene, #319-90211, Tokyo, Japan) according to the attached instructions. Complementary DNA (cDNA) was synthesized using ReverTra Ace qPCR RT Master Mix with a gDNA remover kit (Toyobo, #FSQ-301, Osaka, Japan). The qPCR analysis was performed using Fast Start Universal SYBR Green Master Mix (Roche Diagnostic Japan, #04887352001, Tokyo, Japan) with an MJ Mini instrument (BioRad, Hercules, CA, USA). All gene-specific mRNA expression values were normalized by the expression level of the GAPDH housekeeping (reference) gene encoding glyceraldehyde-3-phosphate dehydrogenase. Quantification of gene expression was performed using ΔCt values, defined as the difference between the target and reference gene Ct values. All primer sequences were as described previously [7].

### 2.6. Quantitative Evaluation of 5-ALA Fluorescence Intensity by Spectroscopic Analysis

In 21 of the 50 patients, 5-ALA fluorescence intensity was quantitatively evaluated by spectroscopic analysis. Patients received oral administration of 5-ALA (Alabel Oral; Nobelpharma Co., #877290, Tokyo, Japan) at 20 mg/kg body weight 3 h before elective craniotomy as part of the routine procedures for fluorescence-guided resection of malignant glioma. The signal intensity of 5-ALA fluorescence from the resected tumor and peritumoral area after tumor resection was measured using an optic fiber probe. We performed spectroscopic analysis of 5-ALA fluorescence intensity using a violet laser diode (VLD) (Alcedo LS-VLD: VLD-EX; SBI Pharmaceuticals Co., Tokyo, Japan) and evaluated fluorescence intensity as the signal intensity of 5-ALA fluorescence (arbitrary units; a.u.).

### 2.7. Statistical Methods

Values are described in mean ± standard deviation. Data were compared using a two-tailed Student’s *t*-test (unpaired). Comparisons of more than two groups were conducted using two-tailed one-way analysis of variance with the Tukey post hoc test. Kaplan–Meier plots were formed to estimate unadjusted time-to-event variables. Spearman’s correlation analysis was performed to examine correlations for non-parametric data. Values of *p* < 0.05 were considered significant. All analyses were performed using Office Excel 2016 software (Microsoft, Redmond, WA, USA) and Easy R (EZR) version 1.54 software (Saitama Medical Center, Jichi Medical University, Saitama, Japan) [18].

## 3. Results

### 3.1. Patient Characteristics

The 50 patients enrolled in this study comprised 37 men and 13 women. The follow-up period was for nine years between April 2014 and March 2023. All patients were newly diagnosed cases receiving primary therapy. The mean age was 65.3 years (range, 19–86 years). No significant differences were found in terms of age or sex. The median KPS score was 80 (range, 60–100). The mean body mass index (BMI) was 21.0 kg/m^2^ (range, 14.5–25.9 kg/m^2^). All patients underwent craniotomy for tumor resection, followed by radiotherapy (60 Gy) and chemotherapy with temozolomide, in accordance with the Stupp protocol [1,2]. The extent of resection was evaluated by volumetric analysis on MRI before and after surgery, as previously described [19]. GTR was achieved in 31 patients (62.0%), STR in 5 patients (10.0%), and PR in 14 patients (28.0%). Histopathological evaluation verified that all tumors were GBM and that 45 tumors lacked mutation in the gene encoding isocitrate dehydrogenase-1 (IDH-1). Five tumors showed mutation in the gene encoding IDH-1. The presence of hotspot mutations in IDH1 (R132) was analyzed by Sanger sequencing. Consequently, 45 tumors corresponded to “GBM, IDH-wild type” and 5 tumors were classified as “astrocytoma IDH-mutant” according to the World Health Organization classification of central nervous system tumors in 2021 [20]. The methylation status of the O6-methylguanine-DNA methyltransferase (MGMT) promoter was analyzed by quantitative methylation-specific PCR after bisulfate modification of genomic DNA, and we used a cutoff of ≥1% for MGMT promoter methylation [21]. Methylation was recognized in the MGMT promotor in 22 cases and mean Ki-67 SI in the core of tumors was 33.3% (range, 7.0–60.0%). The characteristics of enrolled patients are summarized in Appendix A.

### 3.2. Classification of GBM Patients to HI and LI Types on MRI

Out of fifty patients, 28 were classified as HI type and 22 were classified as LI type based on MRI. Met-PET was performed to confirm invasiveness in patients for whom the invasive type could not be determined based on MRI findings alone. The follow-up period was for nine years between April 2014 and March 2023. No significant differences were seen between HI and LI types of GBM in terms of mean age (64.5 vs. 66.4 years, respectively); sex (M/F: 23/5 vs. 14/8, respectively); median KPS score (70 vs. 80, respectively); and Ki-67 SI (33.5 ± 17.2% vs. 31.9 ± 17.4%, respectively). In terms of tumor resection, resection rate ≥95% (GTR or STR) was achieved for 64.3% of HI-GBM and 81.8% for LI-GBM, representing no significant difference. Methylation status of the MGMT promotor and mutation of IDH1 did not differ significantly between HI- and LI-GBMs (Table 2). Survival was compared between patients with HI-type and LI-type GBM. Both progression-free survival (PFS) and overall survival (OS) were much longer in patients with LI-type GBM than in those with HI-type GBM (median PFS: HI 6.7 months, LI 10.8 months, *p* = 0.017; median OS: HI 15 months, LI 28 months, *p* < 0.001) (Figure 2).

Invasive type was determined by specific features on MRI. Both progression-free survival (PFS) and overall survival (OS) were significantly longer in patients with an LI-type GBM than in those with HI-type GBM (median PFS: LI vs. HI, 10.8 months (M) vs. 6.7 months, *p* = 0.017; median OS: LI vs. HI, 28.0 months vs. 15.0 months, *p* < 0.001). Inv: Invasive.

### 3.3. Expression of CD44 mRNA in the Core and Periphery of GBM and CD44 Expression as Evaluated by P/C ratio in HI- and LI-Type GBM

Expression of CD44 mRNA tended to be lower in the tumor core of HI-type GBM than in LI-type GBM, but no significant difference was seen between types (Figure 3a). In contrast, both CD44 expression in the tumor periphery and P/C ratio were significantly higher in HI-type GBM than in LI-type GBM (Figure 3b,c). In addition, the P/C ratio of CD44 expression showed much higher sensitivity and specificity for predicting invasiveness of GBM (100% and 83.3%, respectively) than CD44 expression in the tumor periphery (85.7% and 79.2%, respectively). Survival was compared between patients with a high-CD44 expression type of GBM and a low-CD44 expression type of GBM. Both PFS and OS were significantly longer in the patients with a low-CD44 expression type of GBM than those with a high-CD44 expression type of GBM (median (m) PFS: Low P/C ratio vs. High P/C ratio, 9.5 months vs. 7.0 months, *p* = 0.02; mOS: Low P/C ratio vs. High P/C ratio, 25.0 months vs. 14.0 months, *p* = 0.0000025) (Figure 4).

### 3.4. Quantitative Analysis of Spectroscopic 5-ALA Fluorescence Intensity in the Invasion Area

In 21 patients, to evaluate the strength of GBM invasion, we applied spectroscopy of 5-ALA fluorescence using a VLD with a wavelength of 400 nm. As the area illuminated by the VLD was small, like a spotlight, fluorescence signals were obtained from several sites in target tissues. Since the spectroscopic signal intensities of 5-ALA fluorescence varied within the same target areas, the intensity of 5-ALA fluorescence was expressed not by individual values but by the range of fluorescence intensities in each target, which were divided into four groups according to the strength of signal intensities. These included groups of strong high fluorescence intensity (HFI-S; ≥5000 a.u.), moderately high intensity (HFI-M; ≥3000 a.u. but <5000 a.u.), weak high intensity (HFI-W; ≥1000 a.u., but <3000 a.u.), and low fluorescence intensity (LFI; <1000 a.u.). These intensity levels of each group corresponded to tumor density represented as the number of proliferative tumor cells (Ki-67 SI) (Figure 5a). The HFI-S group showed a large number of tumor cells with a mean Ki-67 SI of 32.8% (range, 20.0–73.3%), the HFI-M group revealed a moderate number of tumor cells with a mean Ki-67 SI of 18.4% (range, 10–40.0%), and the HFI-W group showed a few infiltrating tumor cells with a mean Ki-67 SI of 1.82% (range, 1.0–3.5%) (Figure 5b; Table 3). With 5-ALA signal intensity <1000 a.u. (LFI), almost no tumor cells were detected in peritumoral tissue, showing a Ki-67 SI of 0.27% (range, 0–1.0%).

### 3.5. Evaluation of Invasiveness of GBM by Spectroscopic Analysis of 5-ALA Fluorescence Intensity

To examine the degree of invasiveness in GBM, after contrast-enhancing tumors were resected, the amount of residual tumor cells infiltrating the peritumoral area was assessed by spectroscopic 5-ALA fluorescence intensity in the tissue of the wall of the resection cavity (Appendix A). The peritumoral area showing a 5-ALA fluorescence intensity ≥5000 a.u. was defined as area “A0”, the area showing an intensity of ≥3000 to <5000 a.u. was “A1”, and the area showing an intensity of ≥1000 to <3000 a.u. was “A2”. The peritumoral area presenting 5-ALA intensity <1000 a.u. was regarded as having almost no tumor cells (Figure 6a). Six tumors showed 5-ALA intensity <1000 a.u. in peritumoral brain tissue at the final resection stage. These tumors were therefore regarded as LI-type GBM. The remaining 15 tumors presenting 5-ALA intensity ≥1000 a.u. at the final resection stage were classified as HI-type GBM (Table 3). The ratios of Ki-67 SI in the “A1” and “A2” areas to those in the A0 area were calculated to allow for comparisons with Ki-67 SI, with Ki-67 SI in the A0 area defined as 1. Then, using the A1/A0 and A2/A0 ratios, attenuation curves for tumor density were created for LI- and HI-type GBM (Figure 6b). Both A1/A0 ratio and A2/A0 ratio of Ki-67 SI showed significant differences between LI- and HI-type tumors (HI vs. LI at A1: 0.548 ± 0.168 vs. 0.717 ± 0.118, *p* = 0.038; HI vs. LI at A2: 0.077 ± 0.04 vs. 0.03 ± 0.01, *p* = 0.012). A cutoff of 0.55 for A1/A0 ratio offered 100% sensitivity and 93.1% specificity, while a cutoff of 0.04 for A2/A0 ratio offered 100% sensitivity and 60% specificity (Figure 6c). These results showed that LI-type GBM had cutoff Ki-67 SI ratios of A1/A0 ratio ≥0.55 and A2/A0 ratio <0.04. When the 21 GBMs were re-classified using these cutoff values, one patient (Patient 10) who had been classified with LI-type GBM based on MRI was reclassified as having HI-type tumor on 5-ALA fluorescence spectroscopy. As a result, 7 patients were classified as having LI-type GBM and 14 patients were classified as having HI-type GBM. The 14 HI-type GBMs were further divided into three subgroups by fluorescence intensity at the final resection stage. Six tumors showing fluorescence intensity ≥1000 a.u. to <3000 a.u. were defined as weakly high-invasive (HIw) tumors, five tumors with intensity ≥3000 a.u. to <5000 a.u. were defined as moderately high-invasive (HIm) tumors, and three tumors with intensity ≥5000 a.u. were defined as strongly high-invasive (HIs) tumors. Patients with low-invasive GBM based on 5-ALA showed significantly longer PFS and OS than patients with high-invasive GBM, including all subgroups. Median PFS was: HIs, 4.0 months; HIm, 9.0 months; HIw, 5.9 months; and LI, 15.7 months (*p* = 0.0014). Median OS was: HIs, 8.3 months; HIm, 17.0 months; HIw, 8.4 months; and LI, 26.5 months (*p* = 0.0068) (Figure 6d).

### 3.6. Relationship between P/C Ratio of CD44 Expression and Spectroscopic Signal Intensity of 5-ALA Fluorescence in the Tumor Periphery of GBM

To examine whether CD44 expression in the tumor periphery offers a useful marker to predict invasiveness of GBM, the relationship between P/C ratio of CD44 expression and tumor invasiveness as evaluated by spectroscopic 5-ALA fluorescence intensity was investigated. Tumor invasiveness as determined by the strength of 5-ALA fluorescence intensity in the peritumoral area at final resection correlated well with the A2/A0 ratio of Ki-67 SI at the area A2, except for the relationship between HIw and HIm (Figure 7a). On the other hand, analysis of the relationship between CD44 expression and invasiveness based on 5-ALA fluorescence revealed that tumors in the HI group with high fluorescence intensity ≥1000 a.u. showed significantly higher P/C ratio of CD44 expression than patients in the LI group with fluorescence intensity <1000 a.u. (Figure 7b). However, among high-invasive groups, P/C ratio of CD44 did not correlate with degree of invasiveness among subgroups (Figure 7b). On the other hand, in all 21 GBM patients, P/C ratio of CD44 expression correlated with the A2/A0 ratio of Ki-67 SI at A2 in the peritumoral area (Figure 7c).

## 4. Discussion

High invasiveness is a characteristic feature of GBM and a major obstacle to surgical eradication, leading to early recurrence of GBM. The development of novel therapeutic methods to regulate the invasion of GBM would greatly contribute to improved outcomes for patients with GBM. To control the high invasiveness of GBM, understanding not only the molecular mechanisms underlying GBM invasion, but also the degree of invasiveness in each patient with GBM is crucial. To date, potential markers for numerically predicting the invasiveness of GBM have not been identified. Our previous studies reported that higher expression of CD44 in the tumor periphery compared to the tumor core in GBM correlates with tumor invasiveness as defined from MRI [7]. CD44 is a multi-functional transmembrane glycoprotein that not only mediates cell adhesion and cell migration, but also transduces various signaling pathways of intra- and intercellular events [11,22], thus activating a variety of cellular processes, such as migration, invasion, proliferation, and metastasis in cancer cells, including in GBM [23]. CD44 is also known as a stem cell marker of many cancer cells [24,25]. The migratory and invasive activities of cancer stem cells are highly enhanced by the high expression of CD44 [26], promoting local tumor recurrence and metastasis.

In the present study, we demonstrated that GBMs expressing CD44 much higher in the tumor periphery than in the tumor core (as represented by a higher P/C ratio) were most strongly associated with HI-type GBM. CD44 expression in the tumor periphery also correlated with HI-type GBM, but both specificity and sensitivity in differentiating HI-type tumors from LI-type tumors were much greater with evaluation by P/C ratio of CD44 expression than by CD44 expression in the tumor periphery alone. The co-existence of differentiated non-stem glioma cells expressing CD44 may be the cause of such differences. Taking the ratio of CD44 expression in the periphery to that in the core (P/C ratio) could more clearly represent the degree of CD44 expression in GSCs present within the invasive front of the tumor periphery, as described previously [7]. The present study may have revealed another reason for the differences in CD44 expression reflected by P/C ratio and levels in the periphery alone. In HI-type GBM, CD44 expression tended to be much lower in the tumor core than in the tumor periphery (Figure 3a). Consequently, evaluation of CD44 expression in the tumor periphery using the P/C ratio may reflect the specificity of CD44 expression in the periphery much better than that of CD44 expression in the tumor periphery alone. Our previous study found that GSCs from tumor tissues in the periphery of HI-type GBMs, presenting a high P/C ratio of CD44, expressed higher CD44 in stem-like cells and much higher invasion than GSC from GBM with a low P/C ratio of CD44 expression [7,8]. These results may indicate that glioma cells, particularly GSCs highly expressing CD44, actually exist in the tumor periphery. This may be a reason why CD44 expression in the tumor periphery is higher in HI-type GBM than in LI-type GBM, even if one’s evaluating by either CD44 expression in the periphery alone or CD44 expression by P/C ratio. The P/C ratios of CD44 expression also correlated negatively with OS in all GBM patients, including both HI and LI types. This indicates that the P/C ratio of CD44 expression may provide an indicator for differentiating between HI- and LI-type GBM based on a cutoff P/C ratio for CD44 expression. However, analyses of CD44 expression to date have not been able to evaluate how extensively tumors invade the peritumoral area. To respond to this question, we investigated whether CD44 expression as represented by P/C ratio actually corresponded with the degree of invasiveness in GBM patients using intraoperative 5-ALA fluorescence spectroscopy.

Accumulating studies have demonstrated that a fair number of infiltrating tumor cells exist in the peritumoral area beyond the contrast-enhanced main tumor mass [7,27]. These infiltrating tumor cells, particularly GSCs, are thought to represent sources for early recurrence of GBM. However, neither the density nor the extent of tumor cells infiltrating into the peritumoral area can be depicted by standard morphological imaging modalities, such as CT and MRI. In contrast, Met-PET has been shown to be useful for delineating the extent of tumor invasion from solid GBM masses [28,29]. We reported that the extent of GBM infiltration could be estimated by Met-PET analysis [5]. In that study, we demonstrated that infiltrating tumor cells expand to the peritumoral area while decreasing the number of proliferative tumor cells in accordance with decreases in the level of tumor-to-contralateral normal brain tissue ratio (TNR) in Met-PET. Compared to the present study, Met-PET displayed a mean Ki-67 SI of 39.3% at SUVmax (TNR > 2.0), 23.6% at the area of TNR 1.4 and 1.85% at the area of TNR 1.2, respectively. Accordingly, areas A0, A1, and A2 as defined in the present study may correspond to areas of TNR > 2.0, 1.4 and 1.2 on Met-PET, respectively. As almost no tumor cells were identified in the area of TNR 1.2 on Met-PET, areas showing a 5-ALA fluorescence intensity signal <1000 a.u. would indicate areas without tumor invasion. These results suggest that spectroscopic analysis of 5-ALA fluorescence intensity may take the place of Met-PET in evaluating the invasiveness of GBM.

FGS, particularly using 5-ALA, has been a common procedure for achieving safe maximal resection of GBM cells [17,30]. FGS enables visualization of tumor cells and effective resection of tumor cells that emit a distinct red fluorescence from PpIX. However, infiltrating tumor cells cannot be always detected by photodynamic visualization using conventional excitation methods. In contrast, 5-ALA fluorescence spectroscopy allows the neurosurgeon to detect infiltrating tumor cells in the invasion area. When infiltrating tumor cells were irradiated with laser light in the target area, tumor cells that had not been detected by conventional light were prompted to emit red fluorescence and simultaneously the fluorescent spectrum was depicted on the monitoring display. The fluorescence intensity of PpIX appeared on the display as a peak intensity at a wavelength of 636 nm. The 5-ALA fluorescence spectroscopy not only enhanced the visualization of tumor cells, but also enabled quantitative analysis of the fluorescence intensities of infiltrating tumor cells in the tumor periphery of GBM.

In the present study, we analyzed the spectroscopic intensity of 5-ALA fluorescence to evaluate the extent of tumor invasiveness in the peritumoral area of GBM. As spectroscopic fluorescence signals were derived from small spots of the target area, signals were obtained from several spots. As a result, 5-ALA fluorescence intensity was presented as a range of intensities because intensity signals from several spots in each target area showed a large degree of variability. To evaluate degrees of tumor invasiveness based on 5-ALA fluorescence intensity, we created attenuation curves for tumor density with intensity represented by Ki-67 SI ratios at areas A1 and A2 compared to area A0 (A1/A0 and A2/A0, respectively). The curves showed two patterns according to HI and LI types. Decreases in the number of tumor cells from area A0 to area A1 were less in LI-type tumors than in HI-type tumors. In contrast, the reduction in the number of tumor cells from area A1 to area A2 was less in HI-type tumors than in LI-type tumors. Consequently, LI tumors tended to show a graph with a sharp decline, whereas most HI-type tumors presented an almost linear graph. In the present study, all HI-type tumors presented a Ki-67 SI ratio at area A2 (A2/A0) ≥0.04. When cutoff values of A1/A0 ratio and A2/A0 ratio of Ki-67 SI were applied to differentiate HI- and LI-type tumors, only one patient was reclassified from LI-type tumor in the classification based on MRI to HI-type tumor based on 5-ALA classification (Patient 10). The remaining 20 patients showed the same invasion types as the classification of tumor invasiveness based on MRI. This indicates that criteria determining invasive types from MRI may be commonly used for showing invasiveness in GBM. As a result, in the classification based on 5-ALA fluorescence intensity, seven patients showed LI type and 14 patents showed HI type. Patients with HI-type tumor were further divided into three groups by the strength of fluorescence intensity at area A2. These included subgroups showing intensity >1000 to ≤3000 a.u. (HIw), >3000 to ≤5000 a.u. (HIm), and >5000 a.u. (HIs). Survival curves of patients with tumor invasiveness determined on 5-ALA fluorescence presented the same patterns as patients classified by MRI—that is, the group with the LI-type tumor on 5-ALA showed much longer PFS and OS than those with the HI-type tumor in any of the three subgroups of HIw, HIm, and HIs. These results indicate that the spectroscopic analysis of 5-ALA fluorescence may be applicable for determining the invasiveness of GBM. Consequently, we investigated the relationship between degree of tumor invasiveness based on 5-ALA intensity and levels of CD44 expression (P/C ratio). Although the correlation between the two types of invasiveness (low or high) and CD44 expression (P/C ratio) as evaluated by 5-ALA spectroscopy was quite similar to that from MRI, no relationship was seen among HI-type tumors. In contrast, Ki-67 SI ratios at area A2 (A2/A0 ratio) revealed a positive correlation with P/C ratios of CD44 expression. These results may indicate that the P/C ratio of CD44 expression could better predict the invasive types of GBM, but the degree to which levels of CD44 expression correspond to the degree of GBM invasiveness was not determined. Although the fluorescence intensity of 5-ALA on spectroscopy could not be evaluated by individual numerical values of intensity, the method clarified that almost no tumor cells exist in the peritumoral area where spectroscopy shows a fluorescence intensity <1000 a.u. These data imply that this area may represent a histological border of invasive GBM, and resection of the tumor up to this area while avoiding impairment of normal brain function allows more effective supra-total resection of GBM. From the present study, CD44 expression (represented by P/C ratio) and spectroscopy of 5-ALA fluorescence were recognized as potential biomarkers for predicting GBM invasiveness. However, these predictive methods have several disadvantages. One is that these methods cannot evaluate tumor invasiveness until the tumor is being resected. Preoperative evaluation of tumor invasiveness must therefore still rely on imaging modalities such as MRI and Met-PET. Another disadvantage is that current evaluation of tumor invasiveness cannot always consider the existence of GSCs infiltrating the peritumoral area. Concerning this problem in terms of CD44 expression, we are analyzing CD44 isoforms specifically expressed in GSC, as many cancer stem cells express specific CD44 isoforms and the isoforms thus represent biomarkers for each cancer [31,32]. As for 5-ALA spectroscopy, we have identified a novel enhancer of 5-ALA fluorescence that increases 5-ALA fluorescence in not only differentiated glioma cells, but also GBM stem-like cells (submitted). In the future, utilizing these methods of evaluation with CD44 and 5-ALA fluorescence spectroscopy may identify more useful markers for quantitatively and conventionally predicting tumor invasiveness in GBM. In addition, essential roles of CD44 in the cellular process of tumor invasion may be clarified and reveal the molecular mechanisms underlying GBM invasion.

This study was conducted by analyzing data from a relatively small cohort of patients. This may reflect the difficulty of enrolling a sufficient number of GBM cases from a single center and achieving adequate laboratory evaluations, including Met-PET, for diagnosis. Extensive analysis with a larger sample size is required to obtain further definitive conclusions.

The present study demonstrates that high CD44 expression in the tumor periphery correlated with HI-type GBM, in which tumor invasiveness was evaluated by the specific features on MRI, whereas CD44 expression in the tumor core tended to be lower in HI-type tumor. Consequently, the ratio of CD44 expression in the tumor periphery to that in the tumor core (P/C ratio) was found to be more effective for differentiating between LI- and HI-type tumors. To clarify the relationship between P/C ratio of CD44 expression and actual invasiveness of GBM, invasion was quantitatively evaluated by spectroscopic analysis of 5-ALA fluorescence. Tumor invasiveness was evaluated from the spectroscopic intensity signal of 5-ALA fluorescence, which was demonstrated to correlate with tumor density as represented by Ki-67 SI. As tumors showing an intensity signal <1000 a.u. at the final resection stage displayed no tumor cells in the area, these were considered LI-type tumors. In contrast, tumors presenting a fluorescent intensity signal ≥1000 a.u. were included as HI-type tumors. The HI-type tumors with tumor invasiveness determined by 5-ALA intensity showed significantly greater P/C ratios of CD44 expression than LI-type tumors. In addition, P/C ratios of CD44 expression in all 21 GBM patients correlated with the strength of 5-ALA fluorescence at final resection, representing the strength of tumor invasiveness. These results indicate that the level of CD44 expression (as represented by P/C ratio) was associated with the degree of actual invasiveness of GBM and could provide a reliable biomarker for predicting GBM invasiveness. These findings are of great importance for surgical planning and may help with decision-making processes in the treatment of GBM.

## Figures and Tables

**Figure 1 biomedicines-11-02369-f001:**
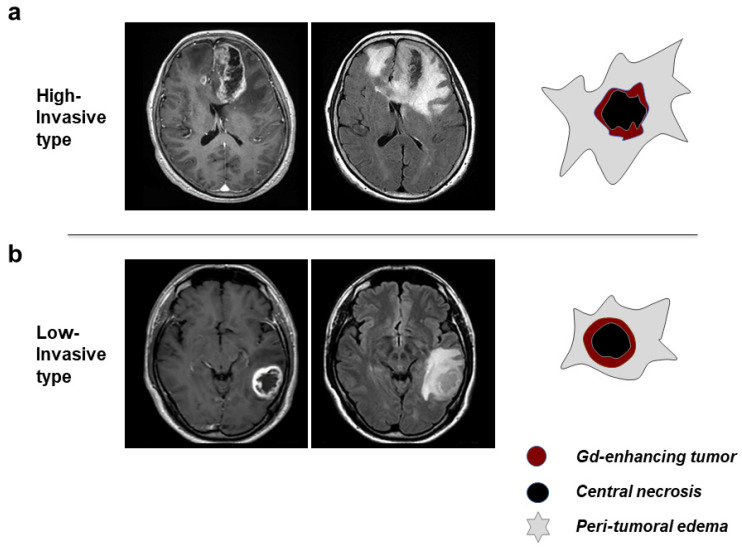
Phenotypes of invasiveness in glioblastoma multiforme (GBM) based on the evaluation of imaging features on magnetic resonance imaging (MRI). GBM was classified into high-invasive (HI) and low-invasive (LI) phenotypes using four features on MRI (Table 1). (**a**) MRI (**left**: gadolinium-enhanced T1-weighted imaging; right: FLAIR) and illustration (**far right**) of HI-type GBM. (**b**) MRI and illustration of LI-type GBM.

**Figure 2 biomedicines-11-02369-f002:**
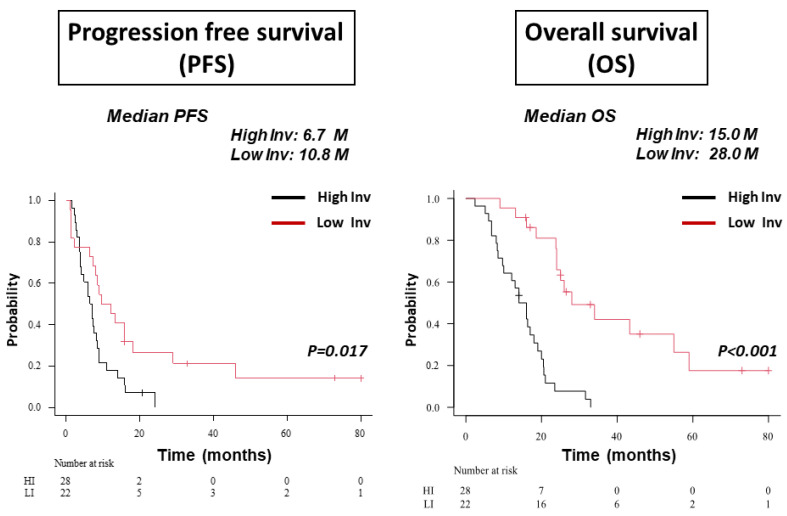
Kaplan–Meier survival curves showing progression-free survival (PFS) and overall survival (OS) in 50 patients with GBM, including HI-type GBM (28 patients) and LI-type GBM (22 patients).

**Figure 3 biomedicines-11-02369-f003:**
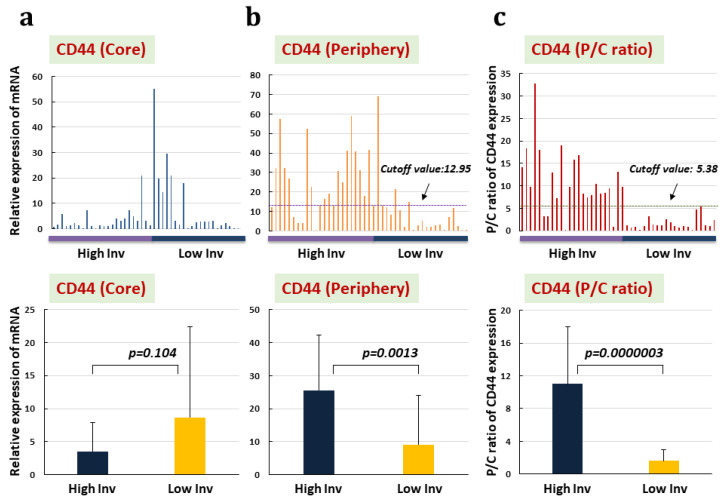
mRNA expression of CD44 in the tumor core and tumor periphery in 50 patients with GBM. Upper panel shows expression patterns of CD44 mRNA in GBM classified into HI-type tumor and LI-type tumor on MRI. Lower panel displays differences in types of GBM invasiveness between HI- and LI-type tumors. (**a**) CD44 expression in tumor core. (**b**) CD44 expression in tumor periphery. (**c**) CD44 expression as represented by P/C ratio (ratio of CD44 expression in tumor periphery to expression in tumor core). No significant difference in CD44 expression in the tumor core is seen between HI- and LI-type tumors (HI vs. LI: 3.55 ± 4.36 vs. 8.67 ± 13.7, *p* = 0.104). In contrast, in the tumor periphery, CD44 expression in HI-type tumors is significantly higher than that in LI-type tumors (HI vs. LI: 25.56 ± 16.83 vs. 9.14 ± 14.93, *p* = 0.0013). P/C ratios of CD44 expression disclose a much greater difference in CD44 expression between HI- and LI-type tumors than CD44 expression in the tumor periphery alone (HI vs. LI: 11.06 ± 7.0 vs. 1.62 ± 1.35, *p* = 0.0000003). These results coincide with sensitivity and specificity for predicting HI-type tumors (P/C ratio: 100% sensitivity, 83.8% specificity; expression in tumor periphery alone: 85.7% sensitivity, 79.2% specificity). (Cutoff values are displayed in CD44 expression patterns in the upper panel.)

**Figure 4 biomedicines-11-02369-f004:**
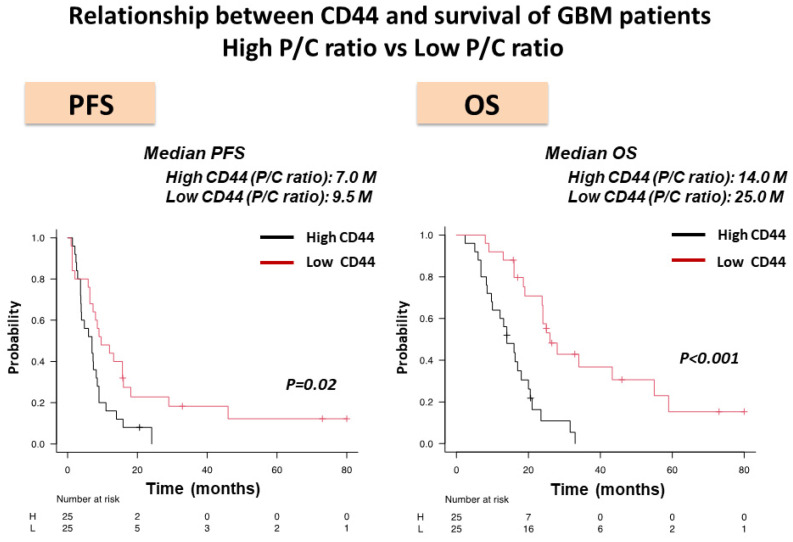
Kaplan–Meier survival curves showing progression-free survival (PFS) and overall survival (OS) in 50 patients with GBM, including GBM with high CD44 expression (High P/C ratio) (25 patients) and GBM with low CD44 expression (Low P/C ratio) (25 patients). CD44 expression types were determined by the cutoff values of 5.38. Both PFS and OS were significantly longer in the patients with a low-CD44 expression type of GBM than those with a high-CD44 expression type of GBM (median (m) PFS: Low P/C ratio vs. High P/C ratio, 9.5 months (M) vs. 7 M, *p* = 0.02; mOS: Low P/C ratio vs. High P/C ratio, 25.0 M vs. 14.0 M, *p* = 0.0000025).

**Figure 5 biomedicines-11-02369-f005:**
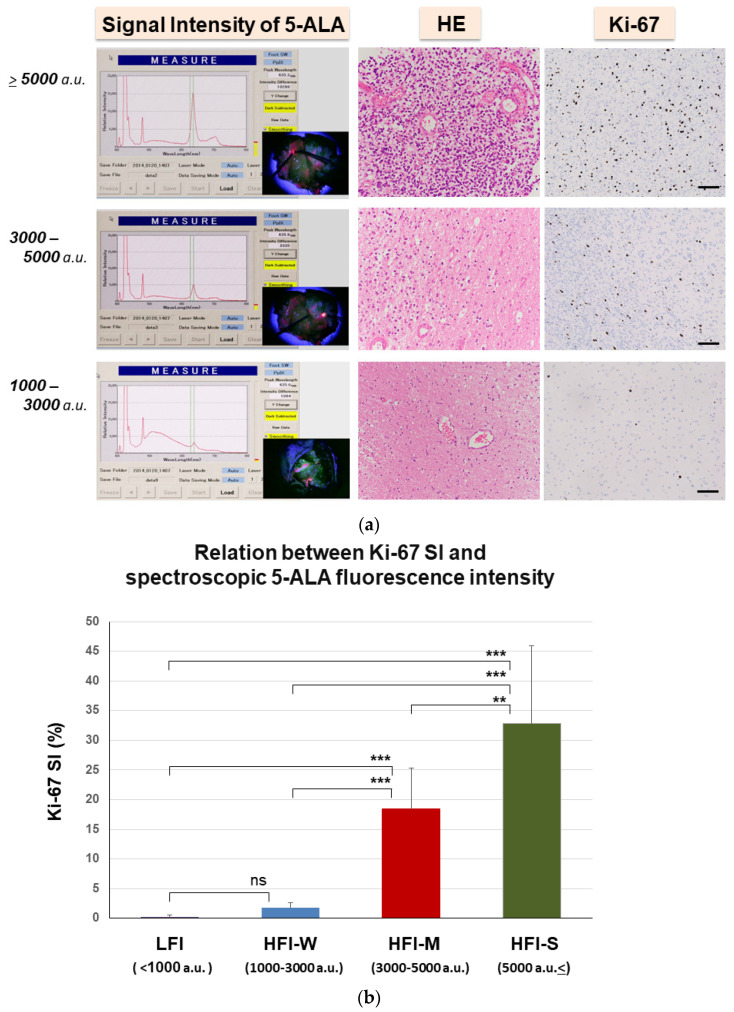
Spectroscopic intensity of 5-ALA fluorescence and its pathology in the target area. (**a**) Left panel: Spectroscopy of 5-ALA fluorescence showing the fluorescence spectrum and fluorescence intensity of the peak at a wavelength of around 636 nm and intraoperative views showing tumors with various shades of red color according to the strength of ALA-fluorescence signal intensity. Right panel: Histological images of hematoxylin and eosin staining and immunohistochemical staining with Ki-67 corresponding to strength of signal intensity shown in the left panels. **Upper**: Tissues at the tumor border show signal intensity ≥5000 a.u. containing numerous tumor cells and a mean Ki-67 LI of 32.8%. **Middle**: Tissues containing medium numbers of tumor cells with a mean Ki-67 SI of 18.4% under a signal intensity of 3000–5000 a.u. **Lower**: Tissues containing few tumor cells with a mean Ki-67 SI of 1.82% under a signal intensity of 1000–3000 a.u. In tissues showing signal intensity <1000 a.u., tumor cells are almost absent. Magnification, ×100. Scale bar, 100 µm. (**b**) A bar graph presenting the relationship between signal intensities of 5-ALA fluorescence and tumor density as represented by Ki-67 SI. The graph demonstrates that intensities of 5-ALA depend on the number of proliferating tumor cells as represented by Ki-67 SI. ** *p* < 0.01, *** *p* < 0.001, ns: not significant.

**Figure 6 biomedicines-11-02369-f006:**
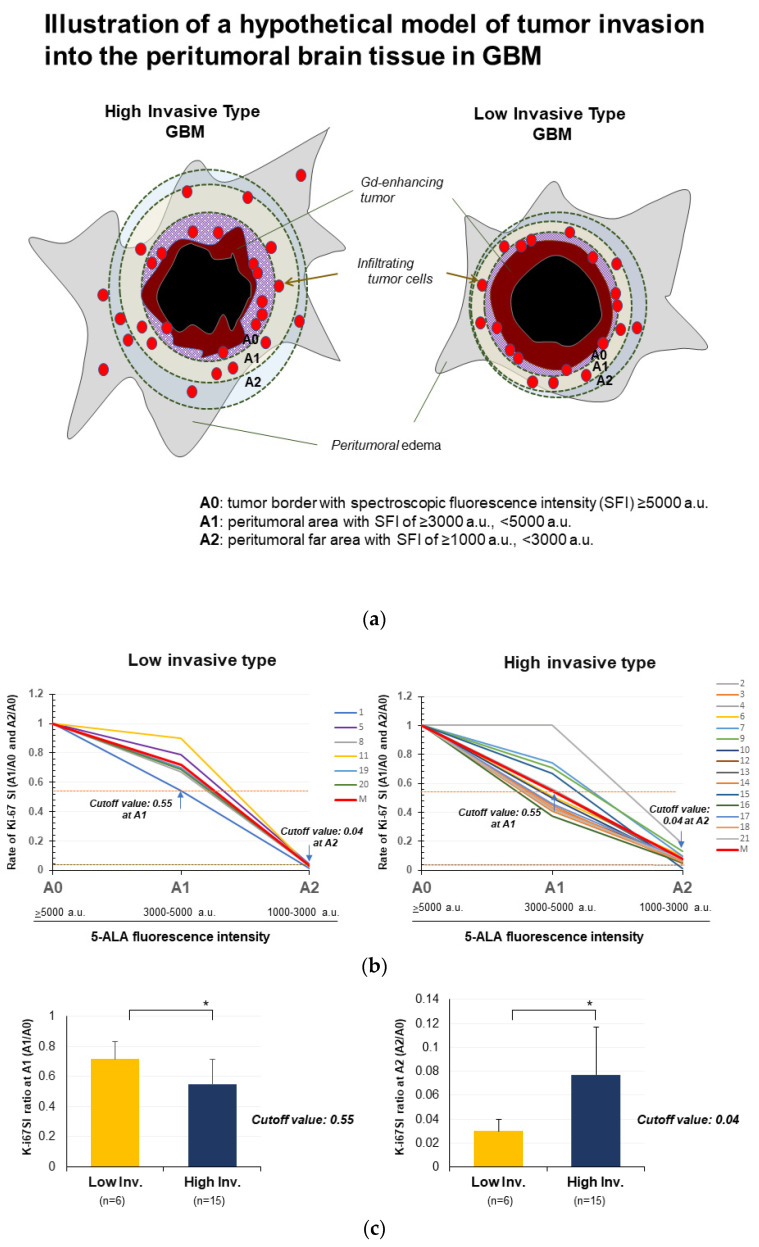
Quantitative evaluation of invasive types in GBM on 5-ALA fluorescence spectroscopy. (**a**) Illustration of a hypothetical model of tumor invasion into the peritumoral brain tissue in HI-type GBM and LI-type GBM. At the tumor border zone, massive numbers of tumor cells crowd the area around the tumor mass. This area is defined as A0. Around area A0, areas are expected to present less signal intensity of 5-ALA than A0. These areas are divided into two areas, with signal intensities of 3000–5000 a.u. (A1) and 1000–3000 a.u. (A2). Tissue samples were obtained from each area and histological examinations including immunostaining of Ki-67 were performed. (**b**) Attenuation curves of Ki-67 SI ratios at A1 and A2 compared to A0 in the group with fluorescence intensity <1000 a.u. at final tumor resection (six patients, corresponding to LI-type GBM) (**left graph**) and in the group with intensity ≥1000 a.u. at final tumor resection (15 patients, corresponding to HI-type GBM) (**right graph**). In LI-type GBM, all six patients show A1/A0 ratios greater than or equal to the cutoff value of 0.55 and A2/A0 ratios less than the cutoff value of 0.04. In HI-type GBM, 14 patients present A2/A0 ratios ≥0.04, except one patient (Patient 15). This patient displays an A1/A0 ratio of 0.67, and so was classified with LI type tumour, the same as the invasive type evaluated on MRI. Although Patient 10 had been classified with LI-type tumor on MRI, both Ki-67 SI ratios of A1/A0 and A2/A0 showed values corresponding to HI-type tumor. (**c**) Bar graphs showing differences in Ki-67 SI ratio at A1 (A1/A0) (**left**) and at A2 (A2/A0) (**right**) between HI-type GBM and LI-type GBM. Values at A1 are significantly higher in LI-type GBM than in HI-type GBM (LI vs. HI: 0.717 ± 0.118 vs. 0.548 ± 0.168, *p* = 0.038). The cutoff of 0.55 offers 100% sensitivity and 60% specificity. Values at A2 are significantly higher in HI-type GBM than in LI-type GBM (HI vs. LI: 0.077 ± 0.04 vs. 0.03 ± 0.01, *p* = 0.0115), with a cutoff of 0.04 offering 100% sensitivity and 93.1% specificity. (**d**) Kaplan–Meier survival curves show PFS and OS in 21 patients with GBM types classified as HI (14 patients, weakly high-invasive (HIw): 6, moderately high-invasive (HIm): 5, strongly high-invasive (HIs): (3) and LI (7 patients), for which invasive types were determined by spectroscopic analysis of 5-ALA fluorescence). Both PFS and OS were significantly longer in patients with an LI-type GBM than in those with HI-type GBM (median PFS: LI vs. HIw, HIm, and HIs: 15.7 months vs. 5.9 months, 9.0 months, and 4.0 months, *p* = 0.0014; median OS: LI vs. HIw, HIm, and HIs: 26.5 months vs. 8.4 months, 17.0 months, and 8.3 months, respectively, *p* = 0.0068). * *p* < 0.05.

**Figure 7 biomedicines-11-02369-f007:**
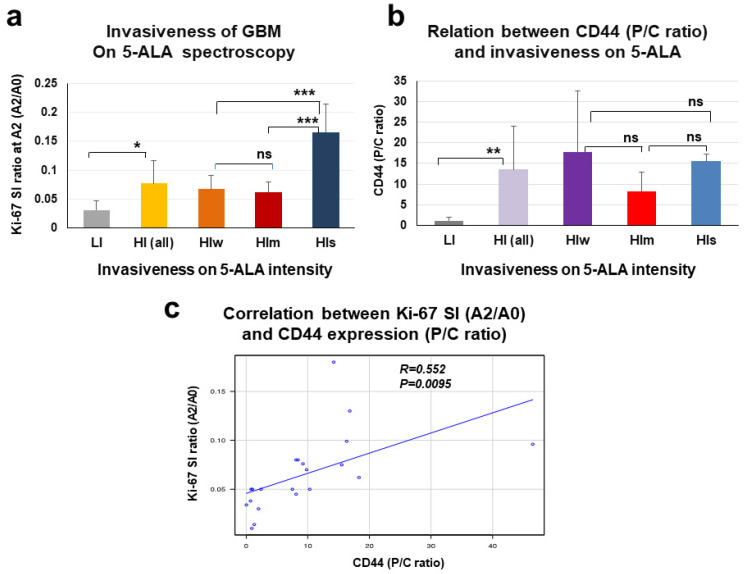
Relation between the degree of tumor invasiveness evaluated by 5-ALA fluorescence intensity and P/C ratio of CD44 expression. (**a**) Relationship between tumor invasiveness measured by 5-ALA fluorescence intensity and tumor density represented by Ki-67 SI ratio at A2 area (A2/A0). LI-type tumors showing fluorescence intensity <1000 a.u. present significantly lower Ki-67 SI ratio than all HI-type tumors showing fluorescence intensity ≥1000 a.u. (LI vs. HI: 0.031± 0.017 vs. 0.077± 0.04, *p* = 0.0115). Among HI-type tumors, HIs (>5000 a.u.) shows significantly higher Ki-67 SI ratio than HIw (1000–3000 a.u.) and HIm (3000–5000 a.u.), but HIw and HIm do not show any difference in Ki-67 SI (A2/A0) (HIs vs. HIw: 0.165 ± 0.05 vs. 0.067 ± 0.024, *p* = 0.00026, HIs vs. HIm: 0.165 ± 0.05 vs. 0.062 ± 0.017, *p* = 0.00014, HIw vs. HIm: 0.067 ± 0.024 vs. 0.062 ± 0.017, *p* = 0.975). (**b**) Relationship between CD44 expression (P/C ratio) and tumor invasiveness measured by 5-ALA fluorescence intensity. HI-type tumors with fluorescence intensity ≥1000 a.u. express significantly higher CD44 than LI-type tumors showing fluorescence intensity <1000 a.u. (HI vs. LI: 13.56 ± 10.59 vs. 1.19 ± 0.8, *p* = 0.0067). However, no significant difference is seen among HI-type tumor groups. (**c**) Spearman’s regression analysis shows a positive correlation between P/C ratio of CD44 expression and Ki-67 SI ratio at A2 (A2/A0). * *p* < 0.05, ** *p* < 0.01, *** *p* < 0.001, ns: not significant.

**Table 1 biomedicines-11-02369-t001:** Criteria for judging invasiveness in GBM based on MRI findings.

Features on MRI	High-Invasive Type	Low-Invasive Type
Morphology of tumor margin	irregular, thin wall	demarcated, thick wall
Enhancement of tumor margin wall	heterogeneous, weak-moderate	homogeneous, intense
Peritumoral brain edema	diffuse, extensive	focal, localized
Morphology of central necrosis	irregular, unevenly located	round, located at or near center

GBM, glioblastoma multiforme. MRI, magnetic resonance imaging.

**Table 2 biomedicines-11-02369-t002:** Patient characteristics in 50 glioblastomas classified as high- and low-invasive type based on MRI.

Tumor Phenotypes	Age (Years)	Sex	KPS (%)	mMGMT	IDH-1 Mut	Ki-67 SI (%)	EOR	
	(Mean ± SD)	Male/Female	(Median)	+/−	+/−	Mean ± SD	GTR + STR ** (%)	PR (%)
High invasive (28 patients)	64.5 ± 13.4	23/5	70	15/12 *	2/26	33.5 ± 17.2	19 (68)	9 (32)
Low invasive (22 patients)	66.4 ± 13.2	14/8	80	7/14 *	3/19	31.9 ± 17.4	18 (82)	4 (18)

MRI, magnetic resonance imaging. SD, standard deviation. KPS, Karnofsky performance status. mMGMT, methylation of the O6-methylguanine-DNA methyltransferase promoter. IDH-1 mut, isocitrate dehydrogenase 1 mutation. MGMT(m), methylation of O6-methylguanine-DNA methyltransferase. Ki-67 SI, Ki-67 staining index. EOR, extent of resection. GTR, gross total resection. STR, subtotal resection. PR, partial resection. * One patient in both phenotypes did not show a definite result. ** Resection more than 95%.

**Table 3 biomedicines-11-02369-t003:** P/C ratio for CD44, Ki-67 SI at peritumoral areas with three different levels of 5-ALA intensity, and 5-ALA intensity at the final tumor resection stage and invasion types of GBM as evaluated by MRI and 5-ALA spectroscopy in 21 patients with GBM.

Patient no.	P/C Ratio	Ki-67 SI (%)			Ki-67 SI Ratio		Fluorescence Intensity (a.u.) at Final Resection (Peritumoral Area)	Type of Invasiveness	
	(CD44)	A0 (≥5000)	A1 (3000–5000)	A2 (1000–3000)	A1/A0	A2/A0		on MRI	on 5-ALA
									High (s)
A1	1.3	73.3	40	<1	0.55	0.014	<1000	L	L
A2	14.2	18.4	18.8	3.5	1.00	0.180	3000–5000	H	H (m)
A3	18.3	44.9	18.6	2.8	0.41	0.062	1000–3000	H	H (w)
A4	9.8	27.1	15.3	2	0.56	0.070	3000–5000	H	H (m)
A5	0.7	26	20.5	<1	0.79	0.038	<1000	L	L
A6	46.5	26.1	13.3	2.5	0.51	0.096	1000–3000	H	H (w)
A7	16.3	23.2	17.2	2.3	0.74	0.099	1000–3000	H	H(w)
A8	2	30	20	<1	0.67	0.030	<1000	L	L
A9	16.8	22.9	16.3	3	0.71	0.130	3000–5000	H	H (m)
A10	0.8	20	10	1	0.50	0.050	3000–5000	L	* H (m)
A11	1	20	18	0.5	0.90	0.025	<1000	L	L
A12	8.1	30	13	2.5	0.43	0.080	3000–5000	H	H (m)
A13	7.5	40	22	2	0.55	0.050	1000–3000	H	H (w)
A14	8.1	44.9	20.2	2	0.45	0.045	1000–3000	H	H (w)
A15	0.9	45	30	0.5	0.67	0.010	≥5000	L	L
A16	10.3	40	15	2	0.38	0.050	1000–3000	H	H (w)
A17	9.2	26	12	2	0.46	0.076	≥5000	H	H (s)
A18	8.4	35	15	2	0.43	0.080	≥5000	H	H (s)
A19	2.4	20	14	0.8	0.70	0.040	<1000	L	L
A20	0.03	48	25	1	0.69	0.034	<1000	L	L
A21	15.5	33	14	2.5	0.42	0.075	≥5000	H	H (s)

P/C ratio, periphery/core ratio. CD44, cluster of differentiation 44. Ki-67 SI, Ki-67 staining index. 5-ALA, 5-aminolevulinic acid. GBM, glioblastoma multiforme. MRI, magnetic resonance imaging. no., number. a.u., arbitrary units. H, high invasive. L, low invasive. H(w), weakly high invasive. H(m), moderately high invasive. H(s), strongly high invasive. * reclassified from L to H on 5-ALA.

## Data Availability

All data used for analysis are presented in the tables in this article.

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
