# Peer review of "Identification of CD44 as a Reliable Biomarker for Glioblastoma Invasion: Based on Magnetic Resonance Imaging and Spectroscopic Analysis of 5-Aminolevulinic Acid Fluorescence"

_biomedicines, 2023, doi:10.3390/biomedicines11092369_

Round 1
Reviewer 1 Report
In glioblastoma multiforme, the P/C ratio of CD44 expression may correlates with GBM invasiveness. To confirm the correlation between CD44 expression and invasiveness of GBM, the present study investigated a larger cohort of patients with GBM and analyzed whether CD44 offered the most reliable indicator for predicting GBM invasiveness. The authors found that CD44 correlated moderately with the proliferation index (ki67), invasive types hat higher CD44 in the periphery (ratio). The prognostic value of CD44 was not analyzed (it is not shown) using K-M and log rank test. The manuscript is well written, it is easy to read and to understand.
Comments:
(1) Line 44. What are the characteristics of the glioma stem-like cells? Are these cells also neoplastic?
(2) Line 50. Could you please show a figure of the CD44 molecule, with the expressing cell, and the receptor, interactions, cell signaling, etc.
(3) What type of cells express CD44 in the tumor periphery?
(4) Line 75. Could you please provide more details regarding that intensity of fluorescence from 5-ALA-derived protoporphyrin IX (PpIX) in brain tumors correlates with tumor cell grade and malignancy?
(5) Ki67 is a proliferation marker. Do all GBM cells express it? Do Ki67 correlates with the histological grade? What are the histological features associated to poor prognosis and/or aggressive invasiveness in GMB?
(6) Line 113. Could you please add that bevacizumab is a monoclonal antibody with VEGF inhibition function (inhibitor)?
(7) Line 116, was the Ki67 analyzed in the center of the tumor, or in the periphery?
(8) In material and methods, could you please add the catalog number of the reagents?
(9) Line 182. Does GSC mean patient-derived glioblastoma stem cell?
(10) Figure 4 shows the correlation of CD44 P/C ratio and the survival of the patients. Using Spearman's bivariate correlation is relatively "unusual". Why not calculating the K-M with log rank test? The p value is significant at 0.037 value, but the R is 0.3, which is a low correlation between the 2 variables, OS and CD44 P/C ratio.
(11) Figure 2 shows a p value of 0.0000005. Please just use p<0.001 (if you agree).
(12) The mRNA expression of CD44 was analyzed in the core and periphery. I understand that the surgical specimen was extracted in peaces. How do you differentiated between the inner and the outer sides? During the macroscopic pathological analysis?
(13) Why did you use mRNA analysis of CD44, when there is immunohistochemistry available? For example, https://www.biosb.com/biosb-products/cd44-antibody-mmab-bsb-12/
(14) Did the cases with IDH-1 mutation have different clinicopathological characteristics?
Author Response
We thank the reviewer for the kind comments. I would like to respond to your comments as follows.
1, The tumor cells are glioma stem-like cells that we established from the primary culture of tumor cells whose tissues were obtained from peritumoral area of GBM. We characterized these glioma stem-like cells and reported in the previous paper (Reference #10, Nishikawa M, et al. Stem Cells Int. 2018). These cells can form tumor spheres in serum-free stem cell medium and have abilities to show multilineage differentiation to neurons and astrocytes, and to make rapid tumorigenesis in mouse brains. As the cells can generate a tumor in mouse brain, they are thought to be neoplastic.
2, As the reviewer pointed out, we present a figure of structure and function of CD44 in the Supplementary Figure S1 in the section of “Introduction” page 2, line 65 in the revised manuscript. In addition, “Supplementary Figure S1” in the past manuscript was changed to “Supplementary Figure S2” as a result of the correction made by this addendum in the section of “Results” page 14, line 458 in the revised manuscript. And, I added the figure legend as follows.
Supplementary Figure S1. Structures and functions of CD44. a) The CD44 gene encodes 20 exons, of which exons 6 to 15 are alternatively spliced and inserted into the variant region of CD44 as variable exons (v1-v10). The figure presents gene structures of CD44 standard isoform (CD44s) and CD44 variant isoforms (CD44v3 and CD44v6). b) Domain structures of CD44 molecule and signaling pathways activated by the interaction of CD44 and hyaluronic acid (HA). CD44 consists of three domains, extracellular domain (ectodomain), transmembrane domain, and cytoplasmic domain. Extracellular domain includes HA binding site and the variant region where variant exons are inserted, thus providing activity as a co-receptor for various growth factors and cytokines. Transmembrane domain activates receptor tyrosine kinase (RTK) and elevates the activities of non-receptor kinases of Src family. The intracellular signaling pathways enhance the activity of the downstream pathways such as mitogen-activated protein kinase (MAPK) and phosphoinositide 3-kinase (PI3K), thus promoting cellular processes including migration, invasion, proliferation, and angiogenesis. Cytoplasmic domain is released by cleavage of transmembrane domain. The released ICD fragments translocate into the nucleus and activate various genes as transcription factor. CD44v6 has a binding site for hepatocyte growth factor (HGF) and vascular endothelial growth factor (VEGF). When VEGF binds to the co-receptor in the variant region, VEGF is enhanced to bind to its receptor VEGFR, resulting in promoting angiogenesis. At this time, activity of c-MET, a receptor for HGF, is inhibited by forming a heterodimer with VEGFR. ERM: Ezrin/Radixin/Moesin.
3, In the tumor periphery of GBMs, CD44 expresses glioma stem-like cells, differentiated glioma cells, astrocytes and neurons (the intensity depends on developmental stage).
4, Fluorescence intensity of 5-ALA-derived PpIX is usually evaluated by visually assessed strength of red color. Tumors showing fluorescence with a strong red color correspond to malignant tumor with high grade (GBM). Tumors presenting almost no red color correspond to low grade glioma, and tumors showing a pale red or pinkish color correspond to grade III malignant glioma. Therefore, we added this content in the revised manuscript “Introduction” page 2, line 88-92.
5, Not all GBM cells show a positive staining for Ki-67. Resting cells that are not in the cell cycle do not express Ki-67. Ki-67 is a potential proliferation marker and the staining index (Ki-67 SI) correlates to the histological grade in all brain tumors including glioma. Histological features for predicting poor prognosis include central necrosis, polymorphism of nuclei, prominent microvascular proliferation, and high mitotic activity.
6, As you mentioned, we added the following sentence as below in the revised manuscript “Materials and Methods” page 4, line 151-152. “bevacizumab, a monoclonal antibody targeting vascular endothelial growth factor (VEGF), therapy at tumor recurrence.”
7, Ki-67 staining was analyzed the tumor cells in the center (core) of GBM using this method.
8, As the reviewer pointed out, we added the catalog number of the reagents in the revised manuscript “Materials and Methods” page 5, line 206-208, page 6, line 216, line 218, line 220 and line 231.
9, This description is my mistake. In the present study, we did not perform the study using GSCs. We eliminated the term “GSC” from the sentence.
10, The reason we used Spearman’s correlation analysis is that we thought it is useful to know whether values of P/C ratio of CD44 expression in each patient with GBM correlates to patients’ survival times. As the reviewer points out, we present the results of a log rank test analyzed by generating Kaplan-Meier survival curves as a new figure by replacing the original Figure 4. The data demonstrated that patients with low CD44 expression (P/C ratio) showed significantly much longer OS and PFS than those with high CD44 expression (P/C ratio). Classification to high CD44 expressing tumors or low CD44 expressing tumors was performed by using a cutoff value of 5.38, whose value was obtained by the study investigating the relation between tumor invasiveness and the level of P/C ratio of CD44 expression. In the revised manuscript “Results” page 9, line 349-354, we presented Kaplan-Meier survival curves with log rank test by replacing to new figures (Figure 4 in the revised manuscript). And, I added the figure legend as follows.
Figure 4. Kaplan-Meier survival curves showing progression-free survival (PFS) and overall survival (OS) in 50 patients with GBM including GBM with high CD44 expression (High P/C ratio) (25 patients) and GBM with low CD44 expression (Low P/C ratio) (25 patients). CD44 expression types were determined by the cutoff values of 5.38. Both PFS and OS were significantly longer in the patients with a low-CD44 expression type of GBM than those with a high-CD44 expression type of GBM (median (m) PFS: Low P/C ratio vs High P/C ratio, 9.5 months (M) vs 7M, p=0.02; mOS: Low P/C ratio vs High P/C ratio, 25.0M vs 14.0M, p=0.0000025).
11, As the reviewer pointed out, we changed the description of p value to p<0.001 in Figure 2 in the revised manuscript “Results” page 8, line 309.
12, We used echo-linked image-guided neuro-navigation devise for intra-axial brain tumors, mainly glioma surgery. As described in our previous reports, we can accurately obtain the tissues at the intended site, including the tissues at the peritumoral border area where non-enhancing tumor cells exist. Please refer to our published papers (Reference #10, Nishikawa M, et al. Stem Cells Int. 2018, Reference #11, Nishikawa M, et al. Trans Oncol. 2021).
13, We understand protein analysis is critical for evaluating CD44 expression. So far, we have investigated expressions of other genes including neural stem cell markers such as Sox2, NANOG, CD133, Oct3/4, and so on at the same time when we measured expression of CD44 mRNA. In the study, we could obtain the data of extensive expressions of such genes in the tumor core and the tumor periphery of GBM tissues. Along the study, we examined both expressions of mRNA and protein of CD44 in several GBM tissues. The results showed that protein and gene expression levels of CD44 in GBM were not so different. From such a viewpoint, we have had measured only mRNA expression of CD44. So, in the present study, we could not analyze the protein expression of CD44 in GBM patients.
14, We confirmed there were almost no difference in survival times and pathological findings between patients with IDH-1 mutated type and those with IDH-1 wild type. In addition, these two groups with different IDH-1 gene status showed no difference in the response to chemotherapy and radiotherapy. They also presented almost the same sensitivity to bevacizumab at tumor recurrence.
Reviewer 2 Report
Inoue and colleagues presented a research article aimed at establishing the prognostic potential of CD44 as a predictive biomarker of glioblastoma invasion. For this purpose, the authors evaluated CD44 expression and peripheral/core ratio in a case series of 50 GBM patients diagnosed with Magnetic Resonance Imaging and spectroscopic analysis using 5-aminolevulinic acid fluorescence. Through different investigations, the authors revealed the good prognostic value of CD44 P/C level alone or in combination with spectroscopic analysis. Overall, the manuscript is interesting, below are suggested some minor revisions that will improve the quality of the manuscript:
1) In the following sentence please avoid the use of the expression “large cohort” since a limited number of patients were recruited: “To confirm the correlation between CD44 expression and invasiveness of GBM, the present study investigated a larger cohort of patients with GBM and analyzed whether CD44 offered the most reliable indicator for predicting GBM invasiveness.”;
2) In chapter 2.4, please indicate the catalog number and dilution of the antibody used;
3) In chapter 3.1, please indicate the follow-up period of the patients enrolled in your study. Same comment in chapter 3.2;
4) Did you evaluate the protein expression of CD44? This information would improve the significance of the results obtained regarding the CD44 mRNA levels;
5) In the Introduction or Discussion section, please emphasize the need for prognostic biomarkers due to the current limitation of GBM treatment. In addition, you should provide a snapshot of the current therapeutic strategies for GBM and how these are often ineffective. For this purpose, please see:
- https://doi.org/10.1016/j.cell.2023.02.038
- https:/doi.org/10.3390/ijms22010351
- https://doi.org/10.3390/cancers14133203
Author Response
We thank the reviewer for the kind comments. I would like to respond to your comments as follows.
1, As the reviewer pointed out, we eliminated “a large cohort” of patients and changed to the following sentence “in an increasing number” of patients in the revised manuscript “Introduction” page 2, line 75.
2, As the reviewer pointed out, we added the catalog number of the reagents in the revised manuscript “Materials and Methods” page 5, line 206-208, page 6, line 216, line 218, line 220 and line 231.
3, As the reviewer pointed out, we added the follow up period as follows “Follow up period was for nine years between April 2014 and March 2023.” in the revised manuscript “Results” page 6, line 250-251 and page 8, line 298-299.
4, As the reviewer comments, analysis of protein expression of CD44 is critical for evaluating CD44 expression. So far, we have measured expressions of various genes including neural stem cell markers such as Sox2, NANOG, CD133, Oct3/4, and growth factors of VEGF and TGF-β, at the same time when measured CD44. Consequently, we extensively analyzed the expression of mRNA of various genes in tumor tissues. In a part of tumor tissues of GBMs, CD44 protein expressions were analyzed by western blot together with its mRNA expression, demonstrating the two expressions showed the almost same results. That is, tumors showing a higher expression of CD44 mRNA in the tumor periphery than in the tumor core presented a higher expression of CD44 protein in the tumor periphery than in the tumor core. Unfortunately, we did not investigate CD44 protein expression in all fifty patients enrolled in the present study.
5, We described the current therapeutic methods for GBM and factors that make the treatment ineffective, and so, the necessity for prognostic biomarkers in the “Introduction”, page 1, line 44-page 2, line 59 of the revised manuscript as showing below.
The current therapy in the primary GBM consists of maximal tumor resection with minimal brain injury, the extended local radiotherapy with 60Gy, and the concomitant chemotherapy with temozolomide. In addition to non-radicality in tumor resection, various factors cause to make GBM extremely poor tumors. These include that GBM cells show a resistance to chemotherapy due to elevated extracellular exclusion of chemotherapeutic agents and the non-disrupted blood-brain barrier, and a resistance to radiotherapy due to upregulation of repair enzymes for DNA damages besides a hypoxic environment surrounding tumor cells. Glioma stem-like cells (GSCs), which constitute a small population in GBM but have abilities to make self-proliferation, multilineage differentiation to neurons and astrocytes, and rapid tumorigenesis in a mouse xenograft model, could participate in all these resistance to the therapy. If GSCs are included in the infiltrating tumor cells, they will become a key cause of early tumor recurrence for GBM [1-6]. Particularly, we found that GSCs expressing high CD44, which were established by the primary culture of tumor cells obtained from the tumor periphery of highly CD44-expressing GBM patients, can present a high invasive feature [10, 11].